# An association sequence suitable for producing ground-state RbCs molecules in optical lattices

Arpita Das[1][*][°], Philip D. Gregory[2][†][°], Tetsu Takekoshi[3], Luke M. Fernley[2], Manuele Landini[1], Jeremy M. Hutson[4], Simon L. Cornish[2] and Hanns-Christoph Nägerl[1]

**1** Institut für Experimentalphysik, Universität Innsbruck, 6020 Innsbruck, Austria
**2** Department of Physics and Joint Quantum Centre (JQC) Durham-Newcastle, Durham University, Durham DH1 3LE, United Kingdom
**3** Alpine Quantum Technologies GmbH, 6020 Innsbruck, Austria
**4** Department of Chemistry and Joint Quantum Centre (JQC) Durham-Newcastle, Durham University, Durham DH1 3LE, United Kingdom

★ arpita.das@uibk.ac.at , † p.d.gregory@durham.ac.uk

## Abstract

We identify a route for the production of $^{87}\text{Rb}^{133}\text{Cs}$ molecules in the $X^1\Sigma^+$ rovibronic ground state that is compatible with efficient mixing of the atoms in optical lattices. We first construct a model for the excited-state structure using constants found by fitting to spectroscopy of the relevant $a^3\Sigma^+ \to b^3\Pi_1$ transitions at 181.5 G and 217.1 G. We then compare the predicted transition dipole moments from this model to those found for the transitions that have been successfully used for STIRAP at 181.5 G. We form molecules by magnetoassociation on a broad interspecies Feshbach resonance at 352.7 G and explore the pattern of Feshbach states near 305 G. This allows us to navigate to a suitable initial state for STIRAP by jumping across an avoided crossing with radiofrequency radiation. We identify suitable transitions for STIRAP at 305 G. We characterize these transitions experimentally and demonstrate STIRAP to a single hyperfine level of the ground state with a one-way efficiency of 85(4)%.

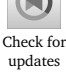
## Contents

° These authors contributed equally to the development of this work.

## 1  Introduction

Arrays of ultracold polar molecules have promising applications for quantum simulation [1–10] and quantum computation [11–17]. Long-range and anisotropic dipole-dipole interactions engineered using dc or ac electric fields allow the exploration of complex many-body Hamiltonians [18]. For single molecules pinned to the sites of an optical lattice, these dipolar interactions are combined with extremely long trap lifetimes. At present, the coldest and densest samples of polar molecules [19–30] are produced in experiments using a two-step indirect method. First, the constituent pre-cooled atoms are associated to form weakly bound molecules by tuning the magnetic field across an interspecies Feshbach resonance. These weakly bound molecules are then transferred to the ground state using stimulated Raman adiabatic passage (STIRAP) [31–33].

$^{87}$Rb$^{133}$Cs was the second polar molecule, after $^{40}$K$^{87}$Rb [21], to be produced in the ultracold regime [22,23]. However, the production of large arrays of RbCs molecules has proved to be difficult. This is primarily due to the scattering properties of the constituent atoms; there is a large background interspecies scattering length ($\sim 650\,a_0$), which renders the atomic clouds immiscible at most magnetic fields. Nevertheless, Reichsöllner *et al.* [34] have demonstrated a protocol for efficient mixing of quantum-degenerate samples of $^{87}$Rb and $^{133}$Cs. First, Bose-Einstein condensates (BECs) of each species are prepared in spatially separated dipole traps. An optical lattice potential is turned on across both samples, with parameters such that Cs crosses the superfluid-to-Mott-insulator transition and Rb remains a superfluid. The magnetic field is then tuned close to a broad interspecies Feshbach resonance at 352.7 G, such that the interspecies scattering length approaches zero. Finally, the Rb is moved to overlap with the Cs, and the trap depth is increased such that Rb also crosses the superfluid-to-Mott-insulator transition. Ideally, this creates an atom array with one Rb and one Cs atom pinned to each site of the optical lattice, from which molecules may be formed efficiently by associating pairs of atoms. So far, lattice filling fractions for double occupancy exceeding 30% have been demonstrated [34].

Previous experiments with $^{87}$Rb$^{133}$Cs have created the molecules by magnetoassociation on a narrow interspecies Feshbach resonance at 197.1 G [22,35]. From here the molecules were transferred to the absolute ground state using STIRAP at 181.5 G with a one-way efficiency of $\sim 90\%$ [22,23,25]. Magnetoassociation following the new mixing protocol is straightforward on the much broader resonance at 352.7 G. However, at this new field the weakly bound states that the molecules can most easily populate are different and therefore the available optical transitions for STIRAP to the singlet ground state are also altered.

In this work, we demonstrate an efficient route for STIRAP to transfer RbCs to the rovibronic ground state at 305 G that is compatible with the protocol for mixing Rb and Cs in optical lattices. We construct a model for the hyperfine structure of the excited state $b^3\Pi_1$, $v' = 29$, $J' = 1$, where $v'$ and $J'$ are vibrational and rotational quantum numbers. We use unprimed, primed and double-primed quantum numbers for the Feshbach, excited and ground states, respectively. We constrain the model using spectroscopy at magnetic fields of 181.5 G and 217.1 G. We combine this with coupled-channel wavefunctions of the weakly bound states to calculate transition dipole moments (TDMs) for the STIRAP transitions previously used at 181.5 G. We present new experiments to characterise the weakly bound states of RbCs around 305 G and use the results to identify suitable STIRAP transitions. We then demonstrate STIRAP to a single hyperfine level of the rovibronic ground state.

The structure of this paper is as follows. In Sec. 2, we describe the near-threshold levels that exist near 181.5 and 305 G and may be used as the starting point for STIRAP. In Sec. 3, we describe the basis set we use to calculate the TDMs. In Sec. 4, we describe our model for the excited state. In Sec. 5, we verify our model by characterising both transitions used for STIRAP at 181.5 G. In Sec. 6, we characterise the weakly bound levels of RbCs near 305 G both experimentally and theoretically. In Sec. 7, we identify suitable STIRAP transitions to the ground state and demonstrate STIRAP experimentally at this new field. Finally, in Sec. 8 we summarize our work and discuss its significance.

## 2  Near-threshold levels

Ground-state RbCs molecules have previously been produced by STIRAP at 181.5 G. Fig. 1(a) shows the weakly bound levels involved near this field, obtained from coupled-channel bound-state calculations [36–39] using the interaction potential of ref. [40]. The states may be labelled by approximate quantum numbers $(n(f_{\text{Rb}}, f_{\text{Cs}})L(m_{f_{\text{Rb}}}, m_{f_{\text{Cs}}}))$; here $f_{\text{Rb}}$ and $f_{\text{Cs}}$ are the total angular momenta of the Rb and Cs atoms, $m_{f_{\text{Rb}}}$ and $m_{f_{\text{Cs}}}$ are their projections onto the quantisation axis provided by the magnetic field $\boldsymbol{B}$, $n$ is a vibrational quantum number, counted down from the energy of the atom-pair state $(f_{\text{Rb}}, m_{f_{\text{Rb}}}) + (f_{\text{Cs}}, m_{f_{\text{Cs}}})$, and $L$ is a quantum number for relative rotation of the atoms. The total parity is $(-1)^L$ and is conserved in a collision, so only states with even values of $L$ can cause resonances in s-wave scattering; values $L = 0$, 2, 4, etc. are indicated by labels s, d, g, etc. Ultracold Rb and Cs atoms are first prepared in their absolute ground states, corresponding to the atom pair state (1,1)+(3,3).

Weakly bound molecules can be created by magnetoassociation on a Feshbach resonance at 197.1 G. The molecules initially enter a very weakly bound state $s1 = (-1(1, 3)s(1, 3))$, known as the least-bound state. This runs almost parallel to the atomic state as a function of magnetic field and is bound by only about 120 kHz. The molecules remain in s1 as the magnetic field is lowered to 182 G, passing over two very narrow avoided crossings with d-wave states on the way which are labelled as (i) and (ii) in Fig. 1(a). At 182 G there is an avoided crossing (iii) between s1 and $d6' = (-6(2, 4)d(2, 4))$. The molecules transfer adiabatically into d6', and then briefly into $d2 = (-2(1, 3)d(0, 3))$ using another avoided crossing (iv), which allows separation of the molecules from the remaining atoms using the Stern-Gerlach effect. The molecules are then transferred back into d6', which is the state used for STIRAP. It is important to note that state d6' has vibrational quantum number $n = -6$; it lies about 16 GHz below the atomic threshold that supports it, so is a shorter-range state than s1 or d2 and this gives it improved Franck-Condon overlap with the excited electronic state used for STIRAP.

When molecules are formed at the Feshbach resonance at 352.7 G, they are again initially in the least-bound state s1 and remain there as the magnetic field is lowered to an avoided crossing near 315 G. We have carried out coupled-channel calculations on the weakly bound states in this region, using a basis set with $L_{\text{max}} = 2$, and the results are shown in Fig. 1(b).

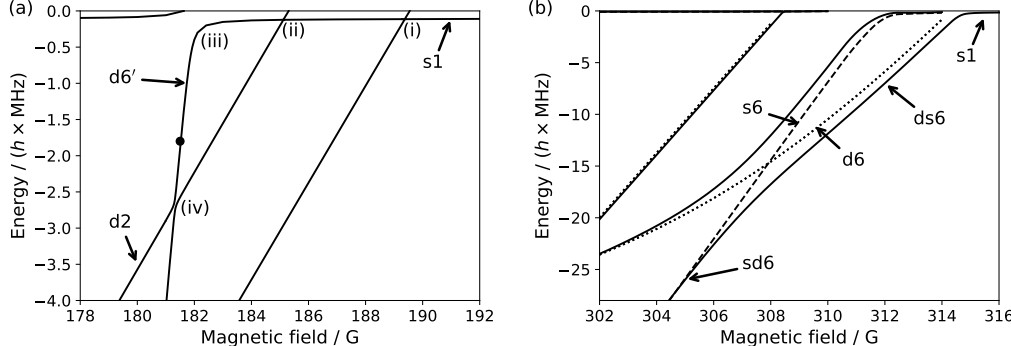

Figure 1: Near-threshold states accessible following magnetoassociation on an inter-species Feshbach resonance at (a) 197 G; (b) 352.7 G. In each case, the molecules are in state s1 between the Feshbach resonance used for association and the maximum magnetic field shown here. The filled circle shows the initial state used for STIRAP at 181.5 G. Avoided crossings in (a) are labelled as described in the main text. In (b), dashed (dotted) lines show states obtained with only s-wave (only d-wave) basis functions.

There are two bound states that undergo avoided crossings with s1 between 310 and 315 G, and the molecules enter the higher-field of these, labelled ds6 in Fig. 1(b). However, in this case the character of the states is strongly dependent on field. There is strong mixing between two underlying states, shown as dashed lines in Fig. 1(b). We designate these underlying states s6 and d6; s6 has character $(-6(2,4)s(1,3))$, while d6 has character $(-6(2,4)d(0,3))$ near threshold but becomes mostly $(-2(1,3)d(1,2))$ at fields below $\sim 305$ G. The dashed and dotted lines on Fig. 1(b) are obtained from coupled-channel calculations that include only s-wave or only d-wave channels, respectively. States s6 and d6 undergo a strong avoided crossing centred near 307 G, at an energy about 14 MHz below the threshold (1,1)+(3,3). The eigenstates, whose energies are shown as solid lines, change character over the avoided crossing; the higher-field state ds6 is mostly of d6 character close to threshold but transitions to the lower-field state sd6 which has dominant s6 character below about 20 MHz. Since s-wave states typically have larger transition intensities than d-wave states to the excited states used for STIRAP, we expect it to be most favourable to perform STIRAP from the deeper part, sd6 in Fig. 1(b). This will be quantified in Sec. 6 below.

Coupled-channel calculations can provide bound-state wavefunctions as well as energies [39,41]. In the present work we perform calculations in a coupled-atom basis set, with basis functions $|f_{\text{Rb}}, m_{f_{\text{Rb}}}; f_{\text{Cs}}, m_{f_{\text{Cs}}}; L, M_L\rangle$. Separate calculations are carried out for each state at each magnetic field of interest. The full coupled-channel wavefunction is expressed as

$$\Psi = R^{-1} \sum_j \Phi_j \psi_j(R), \tag{1}$$

where $R$ is the internuclear distance, $\Phi_j$ is one of the basis functions above, and $j$ is a collective index representing $f_{\text{Rb}}, m_{f_{\text{Rb}}}, f_{\text{Cs}}, m_{f_{\text{Cs}}}, L, M_L$. There is a separate radial channel function $\psi_j(R)$, expressed pointwise on a grid of $R$, for each basis function $j$. The quantum numbers used to identify states are obtained by inspecting the wavefunctions expressed in this basis set, and the wavefunctions themselves are used in the calculations of TDMs described in Secs. 5 and 6 below.

# 3 Choice of basis set for calculating transition dipole moments

For efficient STIRAP we must identify two strong transitions that couple the state F of the Feshbach molecule to the rovibrational singlet ground state G, which has $X\,^1\Sigma^+$ character. The state F has mostly $a\,^3\Sigma^+$ character, because all the contributing states have relatively high spin projections, $M_F = m_{f_{Rb}} + m_{f_{Cs}} \geq 3$. Transitions between pure singlet and triplet states are forbidden, so we exploit the singlet-triplet mixing between the electronically excited states $A\,^1\Sigma^+$ and $b\,^3\Pi$ to allow efficient optical transfer between the initial and final states. Specifically we target the intermediate state $E = (b\,^3\Pi_1, v' = 29, J' = 1)$, which has a small admixture of $A\,^1\Sigma^+$ [42] and was used successfully for STIRAP of RbCs at 181.5 G. We refer to the transitions that connect to $a\,^3\Sigma^+$ as the 'pump' transitions and those that connect to the singlet ground state G as the 'Stokes' transitions, as shown in Fig. 2(a). At 181.5 G, the transitions for STIRAP were found starting from a model without hyperfine structure, and so required an exhaustive search through the many available transitions by experiment [42]. Here we identify suitable transitions by first constructing a model for the electronically excited state, including hyperfine structure. This is used to calculate the relevant energies and, together with the wavefunctions describing states F and G, the TDMs for the candidate transitions.

The system $A\,^1\Sigma^+ - b\,^3\Pi$ has previously been investigated in many different alkali dimers [43–52]. In the case of RbCs, the spin-orbit interaction is large enough that the ratio of the fine-structure splitting $A$ (between the $b\,^3\Pi_0$, $b\,^3\Pi_1$ and $b\,^3\Pi_2$ levels) to the rotational energy $B_{v'}J'(J' + 1)$ is very high ($A/B_{v'} \approx 6300$ for small $J'$) [47]. This makes Hund's case (a) a good description for these states. However, there is no good description of the coupling of the nuclear spins to the other angular momenta at the magnetic fields where we can produce Feshbach molecules. While the physical result does not depend on the basis used for the Hamiltonian matrix, we wish to choose a basis diagonal in most relevant quantum numbers in order to simplify the description of the intermediate state. In addition, we wish to choose a basis in which the initial state F and the final state G can be expressed simply, as the selection rules apply only between molecular states expressed in the same basis set.

For these reasons, we choose to express the wavefunctions for RbCs in terms of Hund's case (a) basis functions with uncoupled nuclear spins,

$$|\Lambda; S, \Sigma; J, \Omega, M_J; i_{Rb}, m_{i_{Rb}}; i_{Cs}, m_{i_{Cs}}\rangle. \tag{2}$$

Here, the quantum numbers $\Lambda$ and $\Sigma$ are the projections of the total electronic orbital angular momentum $L_o$ and spin angular momentum $S$ onto the internuclear axis, with sum $\Omega = \Lambda + \Sigma$. The quantity $J$ is the total angular momentum, including rotation of the molecule, with projection along the quantisation axis $M_J$, which we choose to be along the direction of the applied magnetic field. The nuclear spins for the component nuclei are described by quantum numbers $i_{Rb}, i_{Cs}$ with corresponding angular momentum projections $m_{i_{Rb}}, m_{i_{Cs}}$. Finally, $M_F = M_J + m_{i_{Rb}} + m_{i_{Cs}}$ is the projection of the total angular momentum including nuclear spin onto the quantisation axis. These basis functions do not include the dependence of the wavefunctions on the internuclear distance $R$, which is handled separately.

The wavefunction of the Feshbach state F is obtained from coupled-channel calculations as described in Sec. 2. The coupled-channel wavefunctions are converted to the Hund's case (a) basis as described in Appendix A. Since the state F is of even parity, it can connect only to intermediate states of odd parity. We therefore express the intermediate state in terms of parity-adapted functions [53, 54],

$$\frac{1}{\sqrt{2}}\Big\{|\Lambda'; S', \Sigma'; J', \Omega', M_J'; i_{Rb}', m_{i_{Rb}}'; i_{Cs}', m_{i_{Cs}}'\rangle$$
$$-(-1)^{J'-S'+s'}|-\Lambda'; S', -\Sigma'; J', -\Omega', M_J'; i_{Rb}', m_{i_{Rb}}'; i_{Cs}', m_{i_{Cs}}'\rangle\Big\}, \tag{3}$$

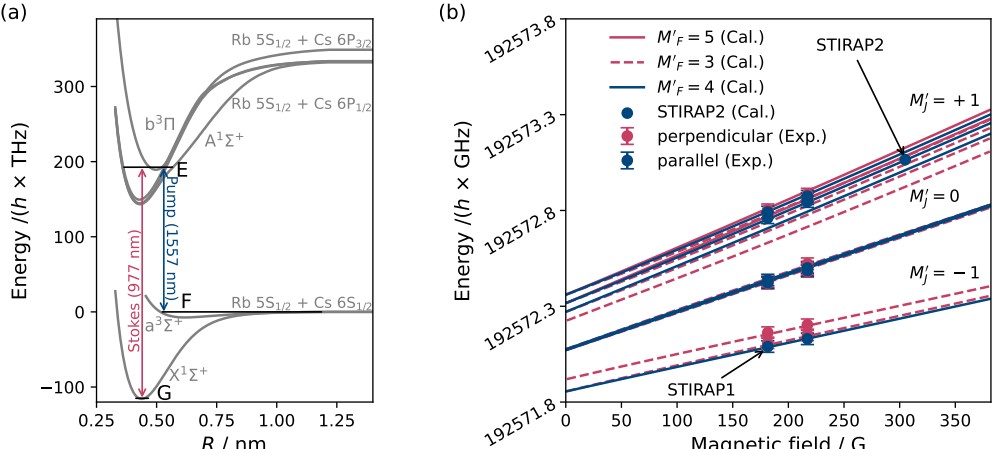

Figure 2: (a) Electronic potential curves for RbCs, showing the STIRAP scheme and corresponding transition wavelengths for our experiment. The initial Feshbach state (mostly $a^3\Sigma^+$), the intermediate (excited) state ($A^1\Sigma^+ - b^3\Pi$, $v' = 29$, $J' = 1$) and the ground state ($X^1\Sigma^+$, $v'' = 0$, $J'' = 0$) are labelled F, E, and G respectively. (b) Zeeman structure of the excited state $E = (b^3\Pi_1, v' = 29, J' = 1)$, which has a small admixture of $A^1\Sigma^+$, with the best-fit parameters from our model fitted to the observed pump transitions at 181.5 G and 217.1 G. The intermediate state labelled 'STIRAP1' was used to perform STIRAP at 181.5 G. The intermediate state used at 305 G is labelled 'STIRAP2'.

where $s'$ is even for $\Sigma^+$ (and higher $\Lambda$) and is odd for $\Sigma^-$. The electric dipole (E1) matrix elements for the pump and Stokes transitions, $\langle E|T_q^1(\vec{\mu})|F\rangle$ and $\langle E|T_q^1(\vec{\mu})|G\rangle$ respectively are given in Appendix A.

## 4 Model for the excited state

To construct a model for the state $E = (b^3\Pi_1, v' = 29, J' = 1)$, we treat the electronic, vibrational, and fine-structure parts of the calculation as already solved. Our Hamiltonian ($H$) consists of three terms to describe the rotational ($H_{\text{rot}}$), Zeeman ($H_Z$), and hyperfine ($H_{\text{hf}}$) structure such that

$$H = H_{\text{rot}} + H_Z + H_{\text{hf}}. \tag{4}$$

Here, the rotational and Zeeman terms are

$$H_{\text{rot}} = B_{v'}\mathbf{J}^2, \tag{5}$$

$$H_Z = g_L\mu_B \mathbf{B} \cdot \mathbf{L}_o + g_S\mu_B \mathbf{B} \cdot \mathbf{S}, \tag{6}$$

where $\mathbf{B}$ is the vector describing the applied magnetic field, $B_{v'}$ is the rotational constant in the excited state, $g_L$ and $g_S$ are $g$-factors associated with the electronic orbital and spin angular momentum, respectively, and $\mu_B$ is the Bohr magneton. The remaining hyperfine term is

$$H_{\text{hf}} = a_{\text{Rb}}\, \mathbf{i}_{\text{Rb}} \cdot \mathbf{L}_o + a_{\text{Cs}}\, \mathbf{i}_{\text{Cs}} \cdot \mathbf{L}_o + \delta\left(b_{f_{\text{Rb}}}\, \mathbf{i}_{\text{Rb}} \cdot \mathbf{S} + b_{f_{\text{Cs}}}\, \mathbf{i}_{\text{Cs}} \cdot \mathbf{S}\right), \tag{7}$$

where the first two terms represent the orbital magnetic dipole interaction and the last two correspond to the Fermi contact interaction. The Fermi contact interaction averages to zero in $b^3\Pi_1$ because the electron spin precesses rapidly around the internuclear axis with no remaining projection ($\Sigma' = 0$). Despite this, there is still a small contribution of the Fermi contact term

due to mixing of $b^3\Pi_0$ with $b^3\Pi_1$ of an amplitude $\delta$, as given in the supplemental material of ref. [47]. We therefore include the Fermi contact term in the hyperfine interaction. Contributions from the electron-nuclear-spin tensor hyperfine interaction and the nuclear electric quadrupole interaction are insignificant compared to the wavemeter measurement uncertainty of 30 MHz and are thus excluded.

The Zeeman terms in the Hamiltonian are off-diagonal in $J'$. We therefore include rotational states up to $J' = 3$ in our model to construct the Hamiltonian, which then produces a $480 \times 480$ matrix representation. By diagonalizing this matrix, we find the energies and wavefunctions of the rotational and hyperfine states.

To constrain our model, we carry out a least-squares fit to the observed pump transitions from one-photon absorption spectra, taken at magnetic fields of 181.5 G and 217.1 G using the apparatus for RbCs molecules at the University of Innsbruck. The Fermi contact constants are fixed at $b_{f_{Rb(Cs)}} = A_{hf_{Rb(Cs)}}/4$ [44], where $A_{hf}$ is the hyperfine coupling constant for the atomic ground state, and the rotational constant is taken from the supplemental material of ref. [47]. There are thus only three fitting parameters: $a_{Rb}$, $a_{Cs}$ and the overall frequency offset. These have best-fit values 154.1(22) MHz $\times h$, 48.7(3) MHz $\times h$, and 192571.564(2) GHz, respectively. The uncertainties in these fitted parameters are due to a combination of the uncertainties in the parameters that are fixed during the fitting and the uncertainty with which the transitions are resolved in the experimental spectra. In principle, for pump light polarised parallel (perpendicular) to the quantisation axis, 6(12) transitions are possible, corresponding to different spin channels $(M'_J, m'_{i_{Rb}}, m'_{i_{Cs}})$, but not all these transitions are resolved in the experiment.

In Fig. 2(b) we show the output of our model of the state E over a broad range of magnetic field from 0 G to 375 G. The red dashed and solid lines represent the states with $M'_F = 3$ and $M'_F = 5$ respectively, and the blue solid lines represent the states with $M'_F = 4$. The red and blue markers with error bars indicate observed transitions F → E for pump light polarised perpendicular (red) and parallel (blue) to the quantisation axis. There are three distinct manifolds of states, which correspond to those with $M'_J = -1, 0, 1$ as labelled.

## 5 Benchmarking the model on the STIRAP transitions at 181.5 G

We first test our model on the transitions previously used for STIRAP at 181.5 G [22,25] as these have been well characterised experimentally. The intermediate state used in this transfer is labelled in Fig. 2(b) as 'STIRAP1'. The initial state $F_{181.5\,G}$ for the transfer is the state d6′ shown in Fig. 1(a), which has character $(-6(2,4)d(2,4))$. To couple the intermediate state with the states $F_{181.5\,G}$ and $G_{181.5\,G}$, the pump light is polarised parallel and the Stokes light is polarised perpendicular to the quantisation axis.

We first calculate the TDMs $\mu_{F_{181.5\,G},E_{181.5\,G}}$ for each pump transition following Eqs. (A.1) and (A.2) in Appendix A. This is then multiplied by a vibrational matrix element, calculated separately for each combination of a component of the Feshbach state with a component of the intermediate state in the case (a) basis. We use vibronic wave functions for the intermediate state from ref. [55] and coupled-channel wavefunctions for the Feshbach state calculated as described in Sec. 2.

The calculated TDMs for all available pump transitions for $F_{181.5\,G} \rightarrow E_{181.5\,G}$ for both parallel (blue) and perpendicular (red) polarisation of the pump beam are shown in Fig. 3(a). It can be seen that two of the transitions have significantly greater TDMs than the others. The transition to state 'STIRAP1' was used in previous studies [22,25]; this state has dominant nuclear spin components $(m'_{i_{Rb}}, m'_{i_{Cs}}) = (3/2, 7/2)$. The transition to this state can be driven with pump light polarised parallel to the quantisation axis. The calculated E1 TDM for this transition is shown in Table (1), along with the measured values from Innsbruck [22], and

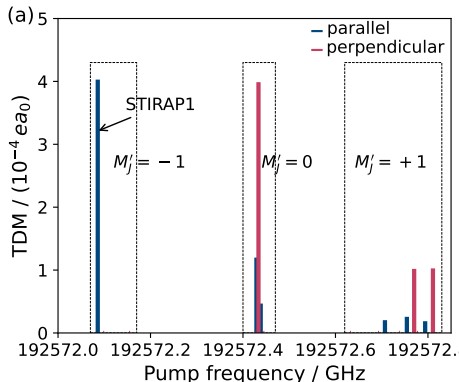 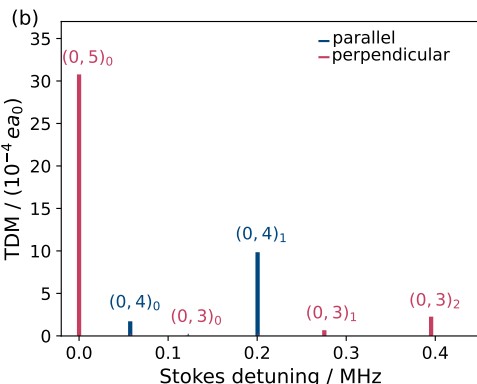

Figure 3: (a) Calculated TDMs for the transitions $F_{181.5\,G} \rightarrow E_{181.5\,G}$ for parallel (blue) and perpendicular (red) polarisation of the pump laser at 181.5 G. The transition frequencies are shown as a manifold of $M_J'$ from left to right. (b) Calculated TDMs for the transitions $E_{181.5\,G}$ ('STIRAP1')$\rightarrow G_{181.5\,G}$ for perpendicular and parallel polarisation of the Stokes laser as a function of Stokes detuning, given with respect to the frequency of the transition to the state with $M_F'' = 5$.

Durham [25]. The values for the Stokes transitions are within about 50% of experiment, but there is roughly a factor of two difference between the calculated and measured values for the pump transitions.

The experimental TDMs are obtained by measuring the Rabi frequency on each transition and normalising it to the intensity of the light. The dominant source of uncertainty in the TDM is from the uncertainty in the intensity of the light. We believe that it is unlikely that these measured values could be incorrect by a factor of 2. Table (1) gives TDMs for the transitions at 181.5 G measured in Durham and Innsbruck and the difference between these values gives a reasonable estimate of the uncertainties present in the experiments.

The differences between the experimental and theoretical values of the TDMs for the pump transitions are probably due to uncertainties in the electronic wavefunctions for the excited states. The calculated TDMs depend strongly on the electronic transition dipole functions, and this dependence is greater for the pump transitions because there is substantial oscillatory cancellation in the radial integrals.

We next calculate the TDMs for the Stokes transitions available from the intermediate state 'STIRAP1'. To calculate the ground-state Zeeman structure we use the Hamiltonian of ref. [56] with the hyperfine constants given in Appendix B. This includes nuclear Zeeman, nuclear quadrupole, nuclear spin-rotation, and scalar and tensor interactions between the nuclear spins. Diamagnetic shifts and ac Stark shifts are neglected. The quantum numbers $N''$ and $M_F''$ are not sufficient to identify all the states uniquely, so we label the states $(N'', M_F'')_k$, where $k$ is an index counting up the states with given $N''$ and $M_F''$ in order of increasing energy.

Table 1: Calculated and measured TDMs for the pump and Stokes transitions at 181.5 G and 305.0 G.

| $|\mathbf{B}|$ / G | Transition | Initial/final state | Excited state | Calculated /$(10^{-4}\,ea_0)$ | Measured /$(10^{-4}\,ea_0)$ | |
|---|---|---|---|---|---|---|
| | | | | | In Innsbruck | In Durham |
| 181.5 | Pump | a $^3\Sigma^+$ ($-6(2,4)$d$(2,4)$) | b $^3\Pi_1$ ($-1, 3/2, 7/2$) | 4.0 | 8(3) [22] | 8.1(1) [25] |
| | Stokes | X $^1\Sigma^+$ $(0,5)_0$ | | 31.0 | 35.0(9) [22] | 28.0(3) [25] |
| 305.0 | Pump | a $^3\Sigma^+$ ($-6(2,4)$s$(1,3)$) | b $^3\Pi_1$ ($+1, 1/2, 5/2$) | 3.1 | —— | 7.2(1) [This work] |
| | Stokes | X $^1\Sigma^+$ $(0,4)_1$ | | 7.8 | —— | 5.1(6) [This work] |

The transition $b^3\Pi_1 \rightarrow X^1\Sigma^+$ is E1 electron-spin forbidden. The Stokes transition is therefore allowed only due to mixing between the singlet $A^1\Sigma^+$ and triplet $b^3\Pi_1$ states caused by the spin-orbit interaction. Supplementary results published by Docenko *et al.* [47] indicate that the state $b^3\Pi_1$, $v' = 29$ used for STIRAP has fractional $A^1\Sigma^+$ character of 0.00029. We calculate the E1 matrix elements for all the possible Stokes transitions $E_{181.5\,G} \rightarrow G_{181.5\,G}$ using Eq. (A.2) in Appendix A. These are combined with vibrational matrix elements as above. The resulting TDMs are shown in Fig. 3(b).

For Stokes light polarised perpendicular to the quantisation axis, there is one strong transition, to the state $(0, 5)_0$; this is the hyperfine ground state at magnetic fields above 90 G and contains only the spin component $(m''_{i_{Rb}}, m''_{i_{Cs}}) = (3/2, 7/2)$. For parallel polarisation, there is also only one strong transition, but to the state $(0, 4)_1$; this is a mixture of the spin components $(1/2, 7/2)$ and $(3/2, 5/2)$. The spin compositions of all accessible states $(N'', M''_F)_k$ are given in Table (3) in Appendix C. STIRAP has been performed successfully to both these ground states [22, 25], but most characterisation has been done using the transition to $(0, 5)_0$. The calculated dipole moment is also given for the Stokes transition in Table (1), along with the measured values from Innsbruck [22] and Durham [25]. In this case, there is reasonable agreement between our calculations and the experimental observations.

# 6 Navigating the near-threshold levels after association at 352.7 G

The largest transition intensities to the excited state targeted for STIRAP are expected for Feshbach molecules prepared in the shortest-range s-wave states. In this section, we perform spectroscopy of the pump transition, and use it to map out the near-threshold bound states that are accessible following magnetoassociation at 352.7 G. We then present our method for preparing the molecules in a state suitable for efficient STIRAP.

The experiments presented from here onwards are performed using the RbCs apparatus at Durham University. We begin with an ultracold mixture of approximately $5 \times 10^5$ $^{87}$Rb and $3 \times 10^5$ $^{133}$Cs atoms in their ground states, $(f_{Rb} = 1, m_{f_{Rb}} = 1)$ and $(f_{Cs} = 3, m_{f_{Cs}} = 3)$, respectively. The magnetic field at the atoms is 21 G. The mixture is confined in an optical dipole trap operating at $\lambda = 1550$ nm, and is levitated by a magnetic field gradient $dB/dz = 32$ G cm$^{-1}$.

To form RbCs molecules, we perform magnetoassociation on an interspecies Feshbach resonance at 352.74 G [40], following a scheme similar to that developed in Innsbruck for the association of Rb and Cs in optical lattices [34]. We jump the magnetic field above the resonance by increasing the magnetic field from 21 G to 355 G in ∼1 ms, and then form molecules by sweeping down across the resonance at a rate of 2.5 G ms$^{-1}$. The molecules are initially formed in state s1, which runs approximately parallel to the free-atom energy. We then ramp the magnetic field down rapidly (∼0.5 ms) to an adjacent Feshbach resonance at 314.74 G [40], where we adiabatically follow an avoided crossing to transfer the molecules into the state ds6 shown in Fig. 4(a); this has principal component $(-6(2, 4)d(0, 3))$. We separate the atoms from the molecules using the Stern-Gerlach effect [20] (discussed in more detail later), and then image the molecules by ramping the magnetic field back up above 353 G, where the molecules are dissociated and the resulting atoms observed with absorption imaging. We form and detect up to 8000 molecules directly after the separation. For spectroscopy and STIRAP we turn up the power of the optical trap over 20 ms and ramp off the magnetic field gradient to transfer the molecules to a purely optical potential. We lose around half the molecules during this procedure.

Light for spectroscopy and STIRAP is derived from a pair of external cavity diode lasers (Toptica DL Pro) locked to a high-finesse ($10^4$) cavity with an ultralow-expansion glass spacer

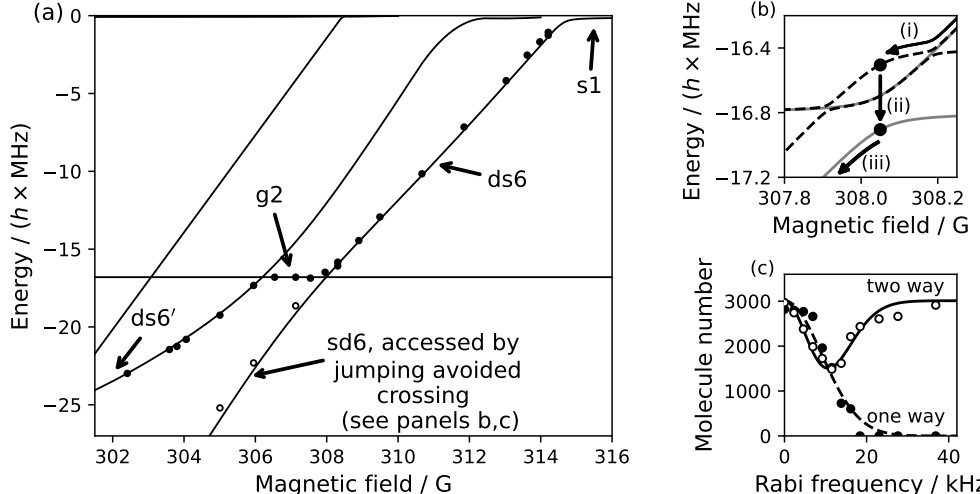

Figure 4: Navigation through the weakly bound states following magnetoassociation at 352.74 G. Directly after formation, molecules occupy the state s1. By decreasing the magnetic field, we access the states labelled ds6, g2 and ds6′ shown in panel (a). Lines show bound-state energies from coupled-channel calculations as described in the text. The markers indicate measurements of the pump transition, with the binding energy inferred assuming a linear Zeeman shift of the excited state. Filled circles show measurements made by simply ramping the magnetic field, whereas empty circles show results obtained after jumping the avoided crossing between ds6 and g2 at 308 G to access sd6. (b) Scheme for transferring molecules to sd6. Dashed (solid) lines show the avoided crossing with (without) a 400 kHz rf field applied. To transfer molecules through the avoided crossing, we follow steps (i-iii) as described in the text. (c) The number of molecules detected after transferring through the avoided crossing once or twice as a function of the Rabi frequency on the rf transition.

as described in ref. [57]. The light is delivered to the molecules in a beam that propagates perpendicular to the magnetic field, and has a waist of 35 $\mu$m at the position of the molecules.

To perform spectroscopy of the pump transition, we expose the molecules to 11 mW of laser light for 500 $\mu$s and measure the number of molecules remaining as a function of the laser frequency. For all experiments shown here, the pump light is linearly polarised parallel to the magnetic field. During the spectroscopy pulse, the optical dipole trap is switched off to avoid ac Stark shifts of the optical transitions, which may vary spatially across the sample. We have previously shown that turning off the dipole trap at $\lambda = 1550$ nm is crucial for efficient STIRAP [25]. We measure the centre frequency of the pump transition as a function of magnetic field, with the molecules initially occupying the state ds6. We measure relative frequency changes in the transition with an uncertainty ($< 160$ kHz) limited by the width of the loss feature. We expect the excited state to shift linearly with magnetic field, so by measuring the pump transition energy we experimentally map out the Feshbach structure as shown by the points in Fig. 4(a). By comparing the measured transition energies to the calculated energies of the near-threshold bound states, we find that the magnetic moment of the excited state is 0.381(6) $\mu_B$; this agrees with the predicted magnetic moment [0.388(4) $\mu_B$] of the state 'STIRAP2', shown in Fig. 2(b), which has $M_J' = +1$ and dominant spin component $(1/2, 5/2)$.

We originally expected that lowering the magnetic field would tune the molecules directly from the d-wave state ds6 to the s-wave state sd6. This was based on the structure shown in Fig. 1, obtained from coupled-channel calculations using a basis set with $L_{max} = 2$. How-

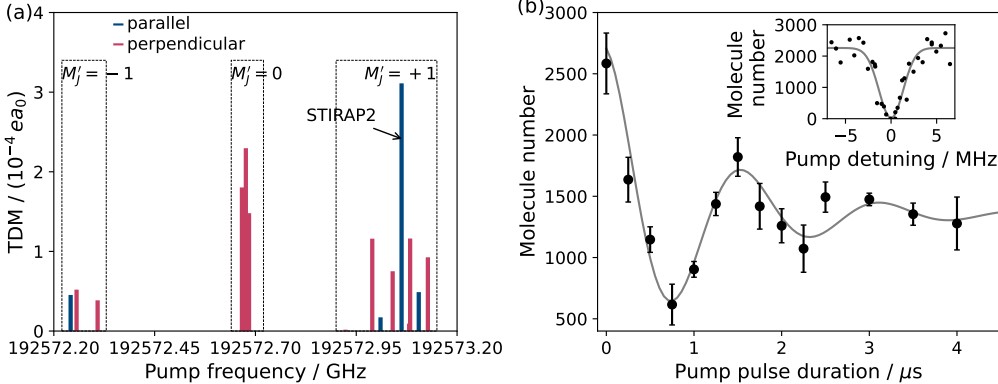

Figure 5: (a) Calculated TDMs as a function of pump frequency for the transition $F_{305\,G} \to E_{305\,G}$ at 305 G with the pump laser polarisation parallel (blue) and perpendicular (red) to the quantisation axis. (b) Characterisation of the pump transition at 305 G. The main panel shows the molecule number as a function of the duration of exposure to resonant pump light. The inset shows the number of molecules remaining in the state sd6 as a function of the pump laser frequency for a 100 $\mu s$ pulse. The line is a Gaussian fit to the results.

ever, we discovered an avoided crossing with an unexpected state that runs almost parallel to threshold at a binding energy near $-17$ MHz. Additional coupled-channel calculations using a basis set with $L_{max} = 4$ identified this as a g-wave state with character $(-2(1,3)g(1,3))$. Because of this, the molecules follow the path shown in Fig. 4 and end up in the d-wave state ds6′. This state has dominant character $(-2(1,3)d(1,2))$ at fields below $\sim 305$ G, as described in Sec. 2.

We use the avoided crossing between ds6 and g2 to facilitate the separation of the atoms and molecules using the Stern-Gerlach effect. In general, this requires that the atoms and molecules possess a different ratio of magnetic moment to mass. However, our current setup can apply only magnetic field gradients that levitate high-field-seeking states (with negative magnetic moment). To perform the separation while keeping the molecules levitated, we therefore set the magnetic field close to the avoided crossing, where we can tune the magnetic moment of the molecules between $+1.1\,\mu_B$ and $-1.4\,\mu_B$.

We calculate the TDM expected for pump transitions from each of the states accessible in Fig. 4(a). We find that only the state $F_{305\,G} = $ sd6, which has dominant component $(-6(2,4)s(1,3))$, can couple strongly to the excited state. The TDM of the pump transitions from the state sd6 $= F_{305\,G}$ to the various hyperfine levels of $E_{305\,G}$ are shown in Fig. 5(a) for parallel (blue) and perpendicular (red) polarisation. The strongest coupling to the excited state is achieved for pump light polarised parallel to the quantisation axis, with the transition expected at a frequency of 192573.1 GHz. The transition to this new state ['STIRAP2' in Fig. 2(b)] has a TDM comparable to that used in the previous STIRAP at 181.5 G.

To enter the state sd6, we must jump over the avoided crossing between the states ds6 and g2. We achieve this using the hybrid transfer method developed by Lang *et al*. [58], as shown in Fig. 4(b). We first tune the magnetic field to 300 mG above the avoided crossing and then switch on a radiofrequency (rf) field at 400 kHz with a Rabi frequency of $\sim 38$ kHz. This is blue-detuned with respect to the width of the avoided crossing between the states ds6 and g2. We then follow a three-step process to complete the transfer: (i) With the rf on, we ramp the magnetic field to the centre of the avoided crossing. This efficiently transfers the population from one side of the avoided crossing to the other by adiabatically following an additional rf-induced avoided crossing between the rf-dressed states. (ii) We switch the rf

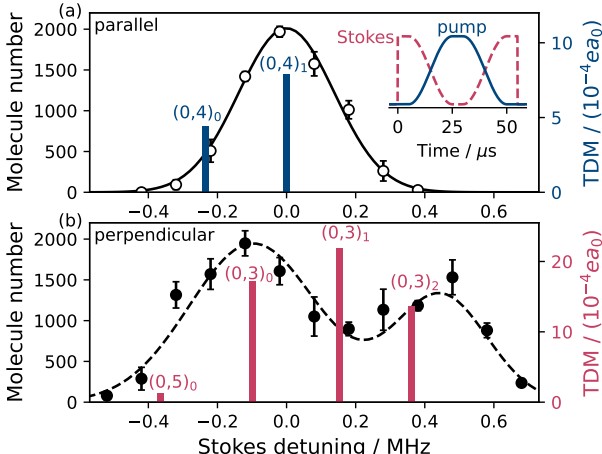

Figure 6: Number of molecules remaining (with empty and filled circles corresponding to the primary $y$-axis) after the round-trip STIRAP pulse shown as inset in (a). We vary the Stokes detuning for light that is linearly polarised (a) parallel and (b) perpendicular to the magnetic field. The line in (a) is a Gaussian fit to the results, with the fitted centre defining zero detuning, which is assumed to correspond to the location of the state $(0,4)_1$. The line in (b) is a sum of two Gaussians fitted to the data to guide the eye. The vertical lines in each plot indicate the accessible hyperfine states of the ground state $X^1\Sigma^+$ ($v'' = 0, N'' = 0$). The height of each line indicates the TDM for that transition corresponding to the secondary $y$-axis.

field off, closing the rf-induced avoided crossing while leaving the molecular state unperturbed. (iii) We continue ramping the magnetic field down, completing the transfer of molecules into sd6. In Fig. 4(c) we show the number of molecules detected after jumping the avoided crossing once and twice. The empty markers in Fig. 4(a) show the binding energy of the molecules inferred from the pump spectroscopy after performing this rf transfer. At 305 G we measure the absolute pump transition frequency to be 192573.4(1) GHz, where the uncertainty is limited by the precision of our wavemeter (Bristol 621A).

We measure the Rabi frequency at which we drive the pump transition for a magnetic field of 305 G, starting from the state sd6. For this, we pulse on the pump light for a variable time, and measure the number of molecules remaining in the Feshbach state as shown in Fig. 5(b). Fitting the oscillation yields a frequency of 632(16) kHz, which corresponds to an intensity-normalised Rabi frequency of 0.8(1) kHz $\sqrt{I_p/(\mathrm{mW\,cm^{-2}})}$. Within uncertainty, this is the same coupling strength as for the transition used for STIRAP at 181.5 G. The measured value of the TDM, derived from the intensity-normalised Rabi frequency, is shown in Table (1) along with the calculated value using our model.

# 7 STIRAP near 305 G

To find the Stokes transition, we apply the STIRAP pulse sequence with peak laser powers of 11 mW for the pump light and 7.6 mW for the Stokes light, with ramp timings optimised for STIRAP at 181.5 G [25]. We first apply the Stokes light for 5 $\mu s$ before a sinusoidal ramp turns the Stokes light off and the pump light on over 20 $\mu s$. After a 5 $\mu s$ hold with the pump light on, the sequence is reversed. When the Stokes light is off-resonant, the pump light removes molecules from the Feshbach state, so we observe a background of no molecules. When the Stokes light is on or near resonance, the population experiences round-trip STIRAP to and

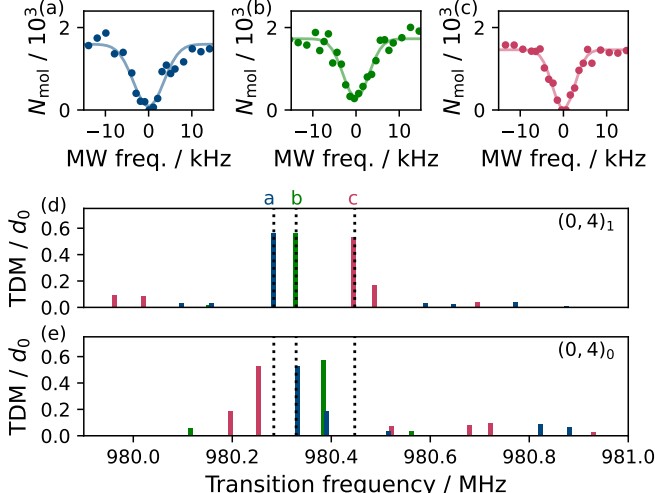

Figure 7: Identification of the state accessed via STIRAP with parallel Stokes polarisation. We perform microwave spectroscopy of the strongest three transitions between $N'' = 0$ and $N'' = 1$, as shown in (a-c). (d,e) show the calculated TDMs in units of the molecule-frame dipole moment ($d_0 = 1.2\,\mathrm{D}$ [22, 23]) for each of the available transitions from either (d) the higher-energy state $(0, 4)_1$ or (e) the lower-energy state $(0, 4)_0$. Blue, red, and green colour codings indicate transitions to states with $N' = 1$ and $M_F'' = 3, 4, 5$ respectively. The centre frequencies of each of the transitions found in (a-c) are indicated by the vertical dotted lines. The microwave spectra observed indicate that the molecules occupy the higher-energy state $(0, 4)_1$.

from the ground state, so the loss is suppressed. We find the Stokes transition to the rotational ground state at a laser frequency of 306831.2(1) GHz. We show the variation in molecule number as a function of the Stokes laser frequency in Fig. 6, for linear polarisation both parallel and perpendicular to the magnetic field. By comparing the initial number of molecules with the number remaining after round-trip STIRAP at the Stokes detunings indicated, we measure maximum one-way efficiencies of 85(4)% and 92(7)% for parallel and perpendicular polarisation, respectively.

Angular momentum selection rules limit the sublevels of the ground state that we can access to those with $M_F'' = 4$ for parallel polarisation and $M_F'' = 3, 5$ for perpendicular polarisation. To identify the states that are populated during STIRAP, we perform microwave spectroscopy [59] of the strongest $\pi$, $\sigma^+$, and $\sigma^-$ transitions from $N'' = 0$ to $N'' = 1$ and compare the results with the hyperfine structure and TDMs calculated using the codes of Ref. [60]. We focus on the state populated with parallel polarisation, as there are only two states with $M_F'' = 4$ that might be accessible. To perform the spectroscopy, the microwave pulse parameters are set to approximate a $\pi$ pulse when close to resonance. The three strongest transitions are shown in Fig. 7 and align well with the transitions expected from the higher-energy state $(0, 4)_1$. Moreover, we find that we can drive Rabi oscillations on each of the available transitions with 100% contrast; this indicates that the molecules occupy just the single hyperfine state $(0, 4)_1$ following STIRAP.

We have calculated the TDMs for the available Stokes transitions from the intermediate state 'STIRAP2' identified in section 6 to all accessible sublevels of the rovibronic ground state. These are shown as the vertical bars in Fig. 6. We see that the detunings where we observe high STIRAP efficiencies broadly agree with the locations of the four strongest transitions. We do not see evidence for STIRAP to the lower-energy state $(0, 4)_0$, even though the predicted TDM

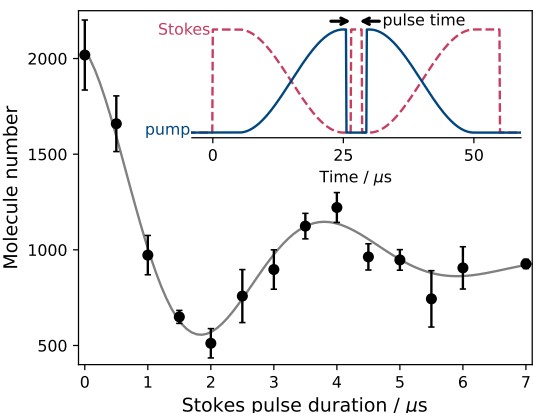

Figure 8: Rabi oscillations on the Stokes transition at 305 G. We pulse on the Stokes light between transfer to the ground state and return to the Feshbach state as shown in the schematic inset. The number of molecules detected after round-trip STIRAP is shown as a function of the duration of the Stokes pulse.

for the transition to this state is substantial. Interference effects that depend on the relative signs of the Stokes matrix elements between nearby transitions can suppress or enhance the STIRAP efficiency [22,61]. The measurements presented here have 100 kHz spacing between detunings, which is probably too broad to resolve the narrow features that interference effects would cause. This may explain the lack of a second peak in STIRAP efficiency at the expected transition to $(0,4)_0$.

To measure the Rabi frequency for the Stokes transition, we pulse on the Stokes light for a variable time between STIRAP pulses. The resulting damped Rabi oscillations are shown in Fig. 8. We extract a Rabi frequency of 250(7) kHz from the oscillations, corresponding to an intensity-normalised Rabi frequency of 0.40(5) kHz $\sqrt{I_{\mathrm{p}}/(\mathrm{mW\,cm^{-2}})}$. This is around a factor of 5 times lower than for the Stokes transition previously used at 181.5 G; the difference arises because the current transition is from a spin sublevel of the intermediate state different from the original. The TDM derived from the intensity-normalised Rabi frequency is shown in Table (1) alongside the calculated TDM corresponding to the state $(0,4)_1$.

## 8   Conclusion

We have found an efficient route to produce $^{87}\mathrm{Rb}^{133}\mathrm{Cs}$ molecules in the rovibronic ground state, compatible with a recently developed protocol for efficient mixing of the atomic species in an optical lattice. To do this, we have constructed a model for the intermediate excited state involved in STIRAP, and used this to calculate TDMs for both pump and Stokes transitions. We have combined this with new calculations and experiments on the weakly bound states of RbCs that exist near 305 G. We encountered an avoided crossing with an unexpected g-wave state at a binding energy near 17 MHz, which interferes with transfer to the s-wave state that is most favourable for STIRAP. We have found a way to jump over this state to reach the target s-wave state. We have demonstrated STIRAP near 305 G, and observed one-way efficiencies of 85(4)% to the $(0,4)_0$ sublevel of the rovibronic ground state at 305 G for parallel polarisation of both pump and Stokes lasers. This is comparable to the efficiency achieved in earlier work with STIRAP at 181.5 G [22,23,25]. Our calculations of TDMs show generally good agreement with the experimental observations, and are able to predict accurately the strongest transitions. This work will allow the production of large, ordered arrays of ultracold polar RbCs molecules.

## Acknowledgements

**Funding information** The Innsbruck team acknowledges funding by the DFG-FWF Forscher-gruppe FOR2247 under the FWF project number I4343-N36, via a Wittgenstein prize grant under project number Z336-N36, and by the European Research Council (ERC) under project number 789017. The Durham authors' work was supported by UK Engineering and Physical Sciences Research Council (EPSRC) Grants EP/P01058X/1, EP/P008275/1 and EP/W00299X/1, UK Research and Innovation (UKRI) Frontier Research Grant EP/X023354/1, the Royal Society and Durham University.

**Data access statement** The data that support the findings of this study are openly available from Zenodo at [62], together with the codes for calculating the Zeeman structure of the excited state and the TDMs. These codes are also available from GitHub at [63].

## A Transformation between coupled-atom basis and Hund's case (a) basis and calculation of E1 matrix element

The transformation between the coupled-atom basis $|f_{\text{Rb}}, m_{f_{\text{Rb}}}; f_{\text{Cs}}, m_{f_{\text{Cs}}}; L, M_L\rangle$ and the Hund's case (a) basis is

$$
\begin{aligned}
&\langle \Lambda; f_{\text{Rb}}, m_{f_{\text{Rb}}}; f_{\text{Cs}}, m_{f_{\text{Cs}}}; L, M_L | \Lambda; S, \Sigma; J, M_J, \Omega; m_{i_{\text{Rb}}}, m_{i_{\text{Cs}}} \rangle = \langle L, \Lambda; S, \Sigma | J, \Omega \rangle \langle L, M_L; S, M_S | J, M_J \rangle \\
&\times \sum_{M_S, m_{s_{\text{Rb}}}, m_{s_{\text{Cs}}}} \langle S, M_S | s_{\text{Rb}}, m_{s_{\text{Rb}}}; s_{\text{Cs}}, m_{s_{\text{Cs}}} \rangle \langle s_{\text{Rb}}, m_{s_{\text{Rb}}}; i_{\text{Rb}}, m_{i_{\text{Rb}}} | f_{\text{Rb}}, m_{f_{\text{Rb}}} \rangle \langle s_{\text{Cs}}, m_{s_{\text{Cs}}}; i_{\text{Cs}}, m_{i_{\text{Cs}}} | f_{\text{Cs}}, m_{f_{\text{Cs}}} \rangle,
\end{aligned}
\tag{A.1}
$$

where $\Lambda = 0$ for the Feshbach state.

The first Clebsch-Gordan coefficient converts from Hund's case (a) to Hund's case (b) [53] and the remainder recouple the electron and nuclear spins.

The E1 matrix elements between case (a) basis functions are

$$
\begin{aligned}
&\langle \Lambda'; S', \Sigma'; J', M_J', \Omega' | T_q^1(\vec{\mu}) | \Lambda; S, \Sigma; J, M_J, \Omega \rangle \\
&= \mu_{\Lambda'\Lambda}^S(R) \delta_{\Sigma',\Sigma} \delta_{S',S} (-1)^{M_J'-\Sigma'-\Lambda'} \sqrt{(2J'+1)(2J+1)} \begin{pmatrix} J' & 1 & J \\ -M_J' & q & M_J \end{pmatrix} \begin{pmatrix} J' & 1 & J \\ -\Omega' & \Lambda'-\Lambda & \Omega \end{pmatrix},
\end{aligned}
\tag{A.2}
$$

and are diagonal in the nuclear spin quantum numbers $m_{i_{\text{Rb}}}$ and $m_{i_{\text{Cs}}}$. Here $\mu_{\Lambda'\Lambda}^S$ is an $R$-dependent electronic transition-dipole matrix element [42]. The delta functions and the two 3-$j$ symbols give the E1 selection rules, with the laser polarisation determining $q = -1, 0, 1$. In our calculations, we consider driving transitions only with light that is linearly polarised parallel ($q = 0$) or perpendicular ($q = \pm 1$) to the quantisation axis.

## B  Molecular constants used in the ground-state calculations

Table 2: The molecular constants [56, 59] used in the ground-state Hamiltonian for $^{87}$Rb$^{133}$Cs.

| | |
|---|---|
| Nuclear spin of Rb ($I_{\text{Rb}}$) | 3/2 |
| Nuclear spin of Cs ($I_{\text{Cs}}$) | 7/2 |
| Nuclear g-factor of Rb ($g_{\text{Rb}}$) | 1.8295 |
| Nuclear g-factor of Cs ($g_{\text{Cs}}$) | 0.7331 |
| Rotational g-factor ($g_{\text{r}}$) | 0.0062 |
| Rotational constant ($B_0$ / MHz) | 490.17 |
| Electric quadrupole coupling constant of Rb ($(eQq)_{\text{Rb}}$ / MHz) | −0.809 |
| Electric quadrupole coupling constant of Cs ($(eQq)_{\text{Cs}}$ / MHz) | 0.059 |
| Nuclear spin-rotation coefficient of Rb ($c_1$ / Hz) | 98.4 |
| Nuclear spin-rotation coefficient of Cs ($c_2$ / Hz) | 194.2 |
| Tensor nuclear spin-spin rotation coefficient ($c_3$ / Hz) | 192.4 |
| Scalar nuclear spin-spin rotation coefficient ($c_4$ / Hz) | 19018.96 |
| Isotropic shielding factor of Rb ($\sigma_{\text{Rb}}$ / ppm) | 3531 |
| Isotropic shielding factor of Cs ($\sigma_{\text{Cs}}$ / ppm) | 6367 |

## C  State compositions of the accessible ground-state sublevels

Table 3: Accessible sublevels $(N'', M_F'')_k$ of the vibronic ground state $\text{X}^1\Sigma^+, v = 0$ at 181.5 G and 305.0 G, with spin components indicated as $|m_{i_{\text{Rb}}}'', m_{i_{\text{Cs}}}''\rangle$.

| Magnetic field / G | Spin state |
|---|---|
| 181.5 | $(0,5)_0 \equiv \|3/2, 7/2\rangle$ |
| | $(0,4)_0 \equiv -0.322\|1/2, 7/2\rangle + 0.947\|3/2, 5/2\rangle$ |
| | $(0,3)_0 \equiv +0.074\|-1/2, 7/2\rangle - 0.366\|1/2, 5/2\rangle + 0.928\|3/2, 3/2\rangle$ |
| | $(0,4)_1 \equiv +0.947\|1/2, 7/2\rangle + 0.322\|3/2, 5/2\rangle$ |
| | $(0,3)_1 \equiv -0.433\|-1/2, 7/2\rangle + 0.826\|1/2, 5/2\rangle + 0.360\|3/2, 3/2\rangle$ |
| | $(0,3)_2 \equiv +0.898\|-1/2, 7/2\rangle + 0.427\|1/2, 5/2\rangle + 0.098\|3/2, 3/2\rangle$ |
| 305.0 | $(0,5)_0 \equiv \|3/2, 7/2\rangle$ |
| | $(0,4)_0 \equiv -0.190\|1/2, 7/2\rangle + 0.982\|3/2, 5/2\rangle$ |
| | $(0,3)_0 \equiv +0.026\|-1/2, 7/2\rangle - 0.228\|1/2, 5/2\rangle + 0.973\|3/2, 3/2\rangle$ |
| | $(0,4)_1 \equiv +0.982\|1/2, 7/2\rangle + 0.190\|3/2, 5/2\rangle$ |
| | $(0,3)_1 \equiv -0.242\|-1/2, 7/2\rangle + 0.943\|1/2, 5/2\rangle + 0.228\|3/2, 3/2\rangle$ |
| | $(0,3)_2 \equiv +0.970\|-1/2, 7/2\rangle + 0.241\|1/2, 5/2\rangle + 0.031\|3/2, 3/2\rangle$ |

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
