# Peer review of "An association sequence suitable for producing ground-state RbCs molecules in optical lattices"

_SciPost Physics, doi:SciPost Phys. 15, 220 (2023)_

## Round 1 · Referee Report · Anonymous (Referee 1) · 2023-8-22

Strengths

  1. Opens a new pathway in an existing or a new research direction, with clear potential for multipronged follow-up work
  2. proposes a "theoretical roadmap" for anyone who is looking for the way to calculate molecular transition strengths relevant for the formation of ultracold molecules, including all angular quantum numbers

Weaknesses

  1. Only minor corrections requested

Report

As summarized by the "strengths" and "weaknesses" above, this paper perfectly meets the criteria to be published in SciPost, in particular, all the "general" criteria are fulfilled. However, I would like the authors to consider the following remarks prior to final acceptance. 1. This paper reports on a very detailed and elaborated theoretical model to fully haracterize the transition strengths of the transitions which would be involved in a STIRAP scheme to create ultracold RbCs molecules. In the way the paper is written, the reader understands that the model has been set up, and has been probed experimentally with the guide of the model. Without minimizing the quality and the usefulness of the model, I wonder if the experimentalists could have "simply" searched for the right transitions, starting from a simple model (namely, without the hyperfine structure) to locate the proper Franck-Condon zone, and then explore one by one the lines composing the hyperfine structure of the excited state, to identify the most suitable one. Could the authors comment a bit in this matter? My reasoning is simple: would the authors recommend to pursue such a theoretical modeling in any case prior to perform the experiment? If yes, I guess that such a "savoir-faire" should be proposed to the community in a way or another (lecture course, code sources) 2. In Section 2, could the authors briefly recall why only even partial waves are considered? 3. I found the description of Figure 1 in Section 2 quite pedagogical and relevant. However, it is sometimes a bit tedious to follow. Would there be a way to include more information in the Figure itself? For instance, would it be possible to number the consecutive avoided croissings, as it is quite tough to identify those which are very weakly avoided? Or may be to add arrrows suggesting the path? In Panel a, the state d6 is not reported. Is it the same than in panel b? Would it be possible to label the unlabellled states, in particular in panel b, for completeness? 4. In Section 3, could the authors recall why the Feshbach resonance is mostly triplet Sigma u? 5. A bibliography issue: the A/b system has been also investigated in K_2 [Eur. Phys. J. D 17, 319{328 (2001)], Li_2 and Na_2 [see references listed in the previous paper] 6. Commenting Table 1 the others wrote that "There is roughly a factor of two discrepancy between the calculated and measured values." As far as I can read, this concerns only the pump transitions. Am I correct? Furthermore, they wrote "our results may indicate that a more accurate electronic wavefunction is required." If my previous remark is correct, how could we explain that the discrepancy occurs for the pump transition and not for the Stokes transition? Moreover, a factor of 2 on the TDM is huge, and usually, electronic wavefunctions are far more accurate to provide values better than a factor of 2. Or would the details of the spin-orbit coupling in the excited state (and thus the respective singlet and triplet fraction of the state) be the main source of uncertainty? I understand that the reason for this is not obvious, but this could be commented in more details, even if I understand that this does not change the conclusions of the paper. Could the experimental procedure to extract the TDM be also uncertain? I did not find in the paper how TDM values are extracted in the experiment, which is usually a quite delicate task.

Requested changes

  1. top of page 3: "We use unprimed, primed and double-primed quantum numbers for the Feshbach, excited and ground states [quantum numbers], respectively."
  2. Figure 2, panel a: the asymptote of the excited state is labeled Rb(5S)+Cs(6P), but there are actually two asymptotes, presumably associated to 6P_1/2 and 6P_3/2. I do understand that the space is limited, but it is an important information to specify that spin-orbit is included in this picture. In this respect the lower index of the Pi state should be clarified (i.e. Omega,defined in the main text)

  • validity: top
  • significance: high
  • originality: high
  • clarity: top
  • formatting: perfect
  • grammar: perfect

Author:  Arpita Das  on 2023-10-25  [id 4061]

(in reply to Report 1 on 2023-08-22)

We thank the reviewer for his/her review of our manuscript by recommending it for publication with more consideration. He/she further posed some valuable questions. Please find the attached response letter, where we have tried to explain all the queries and mentioned the corresponding changes to our revised manuscript.

Attachment:

Response_letter_ArpitaDas.pdf

Author:  Arpita Das  on 2023-10-31  [id 4081]

(in reply to Arpita Das on 2023-10-25 [id 4061])
Category:
remark

I realised the Github link was not included in the previous response letter, and I apologise for this. Please consider the corrected response letter, which is attached here.

Attachment:

Response_letter_ArpitaDas_U2bUYQM.pdf

Author:  Arpita Das  on 2023-10-31  [id 4080]

(in reply to Arpita Das on 2023-10-25 [id 4061])
Category:
remark

I realised the Github link was not included in the previous response letter, and I apologise for this. Please consider the corrected response letter, which is attached here.

Attachment:

Response_letter_ArpitaDas.pdf

---

## Round 1 · Referee Report · Anonymous (Referee 2) · 2023-10-2

Strengths

  1. It provides a comprehensive study of an efficient pathway for producing the rovibrational ground state, applicable to other species of bialkali molecules.
  2. This work opens a new direction in existing research.

Weaknesses

none

Report

The paper presents a comprehensive study of an efficient pathway for producing the rovibrational ground state of RbCs. This investigation focuses on a specific magnetic field of 305 G, at which the formation of a large, ordered array of molecules in optical lattices is anticipated. The paper provides a detailed theoretical model to characterize the weakly bound states, and the model is experimentally verified. Guided by this theoretical model, the authors successfully demonstrate a new STIRAP pathway at 305 G, achieving a competitive transfer efficiency compared to the previous pathway at a lower magnetic field.

The paper is written clearly, and the quality of work is high. Given that this work opens a new pathway in the existing research direction, it meets the acceptance criteria, and I highly recommend its publication.

I have only two small questions:

  1. In the introduction, it states, "Magnetoassociation following the new mixing protocol is straightforward on the much broader resonance at 352.7 G." Is this because the interspecies scattering length is zero at this field, or are there any other reasons? If there are other reasons, it might be helpful to elaborate more.

  2. In section 7 on page 13, the manuscript mentions that the observed 100% Rabi contrast during microwave spectroscopy indicates a single hyperfine state occupancy. Could it still be possible that there is some occupancy of the (0,4)_0 state, and this state also couples to a higher rotational state at a similar microwave frequency with a Rabi coupling similar to the (0,4)_1 to (1,5)_2 transition? I am wondering whether observing a 100% contrast is sufficient to claim single hyperfine state occupancy, although it is highly likely.

  • validity: high
  • significance: high
  • originality: good
  • clarity: high
  • formatting: excellent
  • grammar: excellent

Author:  Arpita Das  on 2023-10-25  [id 4062]

(in reply to Report 2 on 2023-10-02)
Category:
answer to question

We thank the reviewer for his/her review of our manuscript by recommending it for publication. He/she further posed a couple of valuable questions. Please find the attached response letter, where we have tried to explain all the queries and mentioned the corresponding changes to our revised manuscript.

Attachment:

Response_letter_ArpitaDas_PA0o44C.pdf

Author:  Arpita Das  on 2023-10-31  [id 4082]

(in reply to Arpita Das on 2023-10-25 [id 4062])
Category:
remark

I realised the Github link was not included in the previous response letter, and I apologise for this. Please consider the corrected response letter, which is attached here.

Attachment:

Response_letter_ArpitaDas_oEQ9HVp.pdf

---

## Round 2 · Author Response

To the Editor,
Scipost Physics.

We thank you for giving us the opportunity to address the reviewers’ recommendations and
resubmit the manuscript with minor revision.

We have tried to respond to all the recommendations made. The answers to the referees’
questions and comments are given below. We have mentioned the text where ever we have made
changes in the revised manuscript.

Furthermore, we find some typographical errors in our submitted version. We have corrected
them in the revised manuscript. We also rerun our code to calculate the TDMs for the Stokes transitions, with the updated values of some of the constants given in Table 2 according to ref 59, and we provide the updated numbers in the revised manuscript.

We hope that after going through the answers, you will not hesitate to accept our manuscript
for publication in your esteemed journal.

Thanking you.
Kind regards,
Arpita Das

---

## Round 2 · List of Changes

To answer the queries of the reviewers, we have made the following changes/additions to the texts

  1. In Section 2, 1st paragraph of page 3 of the revised manuscript, we have added: " The total parity is $(-1)^L$ and is conserved in a collision, so only states with even values of L can cause resonances in s-wave scattering; scattering; values L = 0, 2, 4, etc. are indicated by labels s, d, g, etc."

  2. In Section 3, 1st paragraph of page 5 of the revised manuscript, we have also changed the texts to "At 181.5 G, the transitions for STIRAP were found starting from a model without hyperfine structure, and so required an exhaustive search through the many available transitions by experiment [42]. Here we identify suitable transitions by first constructing a model for the electronically excited state, including hyperfine structure. This is used to calculate the relevant energies and, together with the wavefunctions describing states F and G, the TDMs for the candidate transitions."

  3. Our submitted manuscript incorrectly mentioned d6 (instead of d6') in the description of the panel (a) in Section 2. It was a typographical error; we have corrected it to d6' in the revised manuscript.

  4. In Section 3, 1st paragraph of page 5 of the revised manuscript, we have added: "The state F has mostly ~a$^3\Sigma^+$ character, because all the contributing states have relatively high spin projections, with $M_F = m_{f_\textrm{Rb}}+m_{f_\textrm{Cs}} \ge 3$."

  5. We have modified the texts at the beginning of the 2nd paragraph of page 5 in Section 3 as “The system A$^1 \Sigma^+$-b$^3 \Pi$ has previously been investigated in many different alkali dimers [43–52].

  6. We have modified the texts in Section 5 as " The values for the Stokes transitions are within about 50% of experiment, but there is roughly a factor of two difference between the calculated and measured values for the pump transitions".

  7. We have added the texts in Section 5: "The differences between the experimental and theoretical values of the TDMs for the pump transitions are probably due to uncertainties in the electronic wavefunctions for the excited states. The calculated TDMs depend strongly on the electronic transition dipole functions, and this dependence is greater for the pump transitions because there is substantial oscillatory cancellation in the radial integrals. "

  8. The labeling of Figure 2 has been modified.

  9. We have changed the texts in Section 7 on Page 13 to read “…we find that we can drive Rabi oscillations on each of these strong transitions with 100\% contrast; this indicates…"

Furthermore, we have found some typographical errors in our submitted version. We have corrected them in the revised manuscript. We also rerun our code to calculate the TDMs for the Stokes transitions, with the updated values of some of the constants given in Table 2 according to ref 59, and we provide the updated numbers in the revised manuscript.

We have also changed 'Transition dipole matrix elements' to 'Transition dipole moments (TDMs)' throughout the texts.

---

## Editorial Decision

published